**Data Availability Statement:** All relevant data are within the manuscript and its Supporting Information files.

**Funding:** The authors were supported by an Alliance for Academic Internal Medicine (AAIM)

# Implementation of an automated scheduling tool improves schedule quality and resident satisfaction

**Frederick M. Howard**[1]*, **Catherine A. Gao**[2], **Christopher Sankey**[3]

**1** Section of Hematology/Oncology, Department of Medicine, University of Chicago, Chicago, Illinois, United States of America, **2** Division of Pulmonary/Critical Care, Department of Medicine, Northwestern University, Chicago, Illinois, United States of America, **3** Department of Internal Medicine, Yale New Haven Hospital, New Haven, Connecticut, United States of America

* fhoward@gmail.com

## Abstract

Rotation schedules for residents must balance individual preferences, compliance with Accreditation Council for Graduate Medical Education guidelines, and institutional staffing requirements. Automation has the potential to improve the consistency and quality of schedules. We designed a novel rotation scheduling tool, the Automated Internal Medicine Scheduler (AIMS), and evaluated schedule quality and resident satisfaction and perceptions of fairness after implementation. We compared schedule uniformity, fulfillment of resident preferences, and conflicting shift assignments for the hand-made 2017–2018 schedule, and the AIMS-generated 2018–2019 schedule. Residents were surveyed in September 2018 to assess perception of schedule quality and fairness. With AIMS, 71/74 (96.0%) interns and 66/82 (80.5%) residents were assigned to their first-choice rotation, a significant increase from the 50/72 (69.4%) interns and 25/82 (30.5%) residents assigned their first-choice in the 2017–2018 academic year. AIMS also yielded significant improvements in the number of night shift/day shift conflicts at the time of rotation switches for interns, with a significant decrease to 0.3 conflicts per intern compared to 0.7 with the prior manual schedule. Twenty-two of 82 residents (27%) completed the survey, and average satisfaction and perception of fairness were 0.7 and 0.9 points higher on a 5-point Likert scale for the AIMS-generated schedule when compared to the non-AIMS schedule. There was no significant difference in the preference for assigned vacation blocks, or in variance for night or ICU rotations. Automated scheduling improved several metrics of schedule quality, as well as resident satisfaction. Future directions include evaluation of the tool in other residency programs and comparison with alternative scheduling algorithms.

## Introduction

Each year, over 30,000 newly graduated physicians begin work within residency programs in the United States as part of the pathway to independent practice [1]. Residency training is

2019 Innovation Grant. The funders had no role in study design, data collection and analysis, decision to publish, or preparation of the manuscript.

**Competing interests:** The authors have declared that no competing interests exist.

comprised of supervised inpatient and ambulatory rotations, with a minimum duration of experience in each practice setting mandated by the Accreditation Council for Graduate Medical Education (ACGME). It is challenging to develop a schedule of rotations that adheres to standards for accreditation, respects intern (first-year trainee) and resident (second- and third-year trainee) preferences, and is perceived as equitable. Schedules must satisfy the staffing requirements of affiliated hospitals, and accommodate trainee vacation preferences and requests for specific rotation experiences. Contingency plans must be developed for illness or other absences, often in the form of a backup or 'jeopardy' rotation–where trainees can be reassigned from non-essential duties to essential ones. The guidelines set forth by the ACGME impose additional constraints, including limitations on total night float (rotations comprised of consecutive night shifts) and intensive care unit rotations, and a minimum amount of time off between shifts. The ideal schedule also minimizes transitions of care, given the potential impact on patient safety [2]. In most internal medicine training programs, chief residents are responsible for scheduling residents for core rotations [3]. Chief residents can spend weeks of time manually designing a schedule, and are limited in the number of constraints they can simultaneously fulfill [4].

The interest in improved approaches to residency schedule is exemplified in a recent review of physician scheduling, in which over one third of identified literature involved resident physicians [5]. Automated systems have been developed to ease the administrative burden of scheduling residents for internal medicine residency programs. Approaches have generally focused on either the scheduling of individual shifts within a set period, or assigning a week- to month-long rotations with pre-set shift schedules to trainees within a year-long schedule. The shift work scheduling problem extends far beyond residency programs, with early model development in the medical field targeted towards nursing personnel [6]. The analogous shift scheduling problem for residency training programs has since been addressed with a variety of methods. Goal programming, with optimization of multiple conflicting constraints, has been used to assign shifts for anesthesia [7] and emergency medicine residents [8]. An integer programming model implemented for a pediatric emergency department successfully improved several important metrics of schedule quality, including suboptimal sleep patterns between shifts and disparity of night shift assignment between residents [9]. This implementation also documented over sixteen hours of time saved in schedule generation per month. Similarly, a spreadsheet implementation of an integer programming model for a radiology residency program increased the minimum interval between call shifts, and residents perceived the computer generated schedule as fairer [10]. The burden on chief residents was not formally measured in this implementation, although the manual schedule generation was described as "rather demotivating", highlighting the importance of alternative approaches.

Whereas the structure of emergency medicine and anesthesia training programs may benefit from optimizing individual shift scheduling, other programs have pre-defined rotations with a set shift schedule which repeats on a weekly to monthly basis. The basic problem of assigning trainees to rotations while satisfying hospital staffing and trainee education requirements is *NP-complete*, requiring exponentially more time to solve when accounting for increasing numbers of trainees and rotations [11]. Therefore, heuristic approaches are often used to identify near-optimal solutions within a reasonable time constraint. The first computerized internal medicine intern and resident schedule was implemented in 1979 at Johns Hopkins, utilizing a "greedy" algorithm to satisfy staffing and educational requirements and vacation requests [12]. Computerized schedules have demonstrated improved ability to satisfy preferences when compared to manual schedules in programs with fewer than 100 trainees [3, 13]. A simplified version of the problem, accounting for staffing and educational requirements without incorporating intern and resident preferences, can be rapidly solved in a "greedy"

fashion for larger programs–this algorithm has been successfully implemented at the University of Illinois College of Medicine [11, 12]. An integer programming model applied to the surgical residency program at Emory University School of Medicine was able to identify near optimal staffing for 28 rotations, although certain requests such as vacation preferences were not accommodated in this solution [14]. Many residency programs have moved towards an "X +Y" block system, where residents alternate between X weeks of inpatient rotations and Y weeks of clinic, and algorithms have been specifically designed to generate schedules for X+Y systems. One implementation at University of Texas Health Center in San Antonio successfully created yearly schedules, but the only quality outcome reported was standardization of clinic capacity [15]. To our knowledge, no open source tool is available for computer-assisted schedule generation in a training program as large as the Yale New Haven Hospital Internal Medicine Residency Program (YNHH-IMRP), for which over 200 trainees are assigned to over 100 individual roles across numerous services. We therefore present a heuristic based scheduling tool, the Automated Internal Medicine Scheduler (AIMS) which was successfully implemented for the YNHH-IMRP during the 2018–2019 academic year and compare metrics of schedule quality before and after implementation.

## Methods

### Design of AIMS tool

AIMS was developed in Microsoft Excel and written with Microsoft Visual Basic for Applications (S1 Application). We collected preference data via online survey (Qualtrics, Provo, UT), in which residents ranked the desirability of vacation timeslots and selected their top three choices for rotations. We also collected free-text comments about scheduling preferences, to account for important life events such as maternity/paternity leave and weddings. Essential scheduling elements, such as the scheduling of vacation during a planned honeymoon, were entered manually into the AIMS tool prior to automated scheduling. Additionally, our program coordinated the scheduling of trainees on internal medicine services from six different residency programs within our institution, as well as external trainees from outside institutions who rotate through Yale services. The number of external trainees from outside programs is variable from block to block and determined by the administration of the outside programs. We do not have the liberty to reschedule these trainees, so they are hard-coded prior to running the AIMS tool. Rotation schedules are generated for a one-year period; for our program, this equates to 26 rotations per trainee, typically 2 weeks in length.

AIMS emulates a sequential scheduling lottery to account for resident preferences, which is commonly used in college coursework, medical student scheduling, and holiday shift scheduling (S1 Fig) [16, 17]. A tally is kept for each resident, initialized at zero, representing how well the schedule fits with the resident preferences. For example, if a trainee receives a first-choice vacation, the tally will decrease by one; if a trainee receives a fourth-choice vacation, the tally decreases by four. Each trainee is assigned two vacations per ACGME requirements, one in the first fourteen blocks, and one in the second twelve blocks of the year–thus, if they were assigned their last choice vacation in the first half of the year, the tally would be decreased by fourteen. Trainees with the lowest tally are then chosen first for the next lottery. After vacation assignments, they will again be chosen by lottery to be assigned a rotation of their preference, if feasible. Our methodology is similar to previous studies, which have utilized a proportion of granted requests to allocate shift assignments equitably [18].

AIMS subsequently iterates through all trainees to assign rotations necessary to satisfy ACGME requirements for emergency medicine and the intensive care unit (ICU). For every rotation with an unfilled two-week block, AIMS iterates through each trainee in a random

order to identify a suitable individual to insert. The constraints applied as trainees are assigned to rotations include: prohibition of sequential night float rotations (minimizing consecutive night float to two-week intervals), limits on the total number of months a trainee can spend on a specific service, limits on total number of intensive care unit and night rotations, and prohibition of rotation assignments with "shift conflicts" (i.e., a night shift on the last day of one rotation, followed by a day shift on the first day of the next rotation) which necessitate utilization of another trainee serving on a backup or jeopardy rotation. Adherence to all constraints does not allow the generation of a complete schedule in some cases, so the process is repeated without the constraint on overlapping clinical responsibilities, as a jeopardy trainee can be called upon to cover these shifts. Once all clinical services are covered, the remaining unscheduled time is converted to elective and jeopardy rotations, based on the year and program of the trainee. A single feasible schedule is thus generated on each run of the program.

Output of the AIMS is displayed in separate Microsoft Excel sheets, one displaying the rotation schedule for each trainee, and one displaying the trainee staffing for each individual rotation. As we use AMiON® scheduling software (Newton, MA) to list daily clinical responsibilities, our program also includes a function to convert block schedules into daily call schedules which can be imported into AMiON.

## Data collection and analysis

Our study was reviewed by the Yale New Haven Hospital Institutional Review Board (Protocol ID 2000024118) and deemed exempt given its educational focus, anonymized survey, and minimal risk nature.

To evaluate the benefits of a schedule generated with AIMS, we compared both objective and subjective metrics of schedule quality across the 2017–2018 and the 2018–2019 academic years, pre- and post-implementation of AIMS. Trainee schedules must satisfy hospital staffing and ACGME requirements, and as such, these were not assessed as comparator measures. We focused on metrics germane to resident satisfaction. Objective measures included accommodation of trainee vacation choice preferences, accommodation of rotation preferences, shift conflicts, and variance in assignment to night, ICU, and jeopardy rotations. These specific schedule metrics were chosen by expert opinion, but have been described in prior studies on trainee scheduling. Accommodation of rotation and vacation preferences has used as a criteria for multiple other scheduling endeavors [3, 19, 20], and cited as a limitation when not included [14]. The difficulty in satisfying vacation and rotation preferences has led to other, non-automated approaches to scheduling [21], highlighting that optimization of these metrics is of high priority and a reasonable markers of schedule quality. Prior to scheduling, Qualtrics surveys of preferred vacation timeframes as well as rotation preferences were collected during the 2017–2018 and the 2018–2019 academic years. This data was abstracted to assess adherence to preferences. To evaluate the fit of vacation preferences, we calculated the average survey rank of the assigned vacation blocks for each trainee, with '1' indicating a resident's top choice. We calculated the number of trainees assigned to their first and second choice rotations, although only first choice preferences were available for interns in the 2017–2018 academic year. These metrics were compared between years using a two-sided unpaired t-tests at the α = 0.05 significance level.

Discordance between overnight call and daytime responsibilities has been previously used to guide schedule creation [20]. We evaluated the number of conflicts between overnight and daytime clinical responsibilities in each schedule, which was again compared using a two-sided t-test. Such conflicts are defined as any night shift or overnight call when the trainee is subsequently scheduled for a day shift on a separate service. Equity of the trainee experience

leads to a perception of fairness, and has been an outcome measure of previous attempts at automation [9]. Night and ICU rotations are less desired due to longer hours and disruptive sleep schedules; conversely, jeopardy rotations may be desired due to more perceived flexibility. We quantify equity by measuring variance in night, ICU, and jeopardy rotations, and we compared variance between the 2017–2018 and 2018–2019 years using the F-test at the α = 0.05 significance level. To ensure that results of these comparisons were not due solely to differing constraints between these two academic years, we regenerated the 2017–2018 schedule using the AIMS tool, which was compared to the 2017–2018 manual schedule.

To assess subjective schedule quality, in September 2018 we surveyed upper-level residents (PGY-2 and PGY-3) who could compare the quality of the current schedule to the previous year's schedule, which was previously created without the use of AIMS or any other scheduling tool. Residents were asked to rate their satisfaction and perceived fairness of both the previous and current schedule on a 5-point Likert scale, using the questions "How satisfied were you with your schedule?" and "How fair was your schedule?" (S1 Text). The results were compared with a paired two-sided t-test at the α = 0.05 significance level. We adjusted for multiplicity of comparisons using the Holm-Bonferroni method. Comparisons for objective metrics of schedule quality and survey results were separately considered for this adjustment as they were based on different sources of data.

Analysis was performed in Microsoft Excel 2013 and Graphpad PRISM Version 7.01.

## Results

We found no significant difference in the number of night, ICU, and Jeopardy rotations for interns, and no significant difference in the variance in assignment of these rotations (Table 1). The average number of night and ICU rotations needing to be filled were similar between the two years (Table A in S2 Text). There were fewer shift conflicts due to overlapping clinical responsibilities for interns after implementation of AIMS (0.7 conflicts per intern, versus 0.3 conflicts per intern after implementation, p < 0.01). With AIMS, 71/74 (96.0%) interns and 66/82 (80.5%) residents were assigned to their first-choice rotation, a significant increase from the 50/72 (69.4%) interns and 25/82 (30.5%) residents assigned their first-choice in the 2017–2018 academic year. For residents, the AIMS schedule demonstrated a lower variance in number of Jeopardy rotation assignments, although this did not remain significant upon adjustment for multiple comparisons (Table 2). There was an increase in variance for ICU rotations in the 2018–2019 year which did not meet statistical significance. Similar to the data for interns, we found a significant increase in the number of residents assigned their first and second choice rotations.

Similar results were obtained when the 2017–2018 schedule was recreated with AIMS (Tables B and C in S2 Text)–with improvements seen in number of shift conflicts, rank of

**Table 1. Intern schedule quality metrics.**

| Mean (SD) | 2017–2018 (n = 72)* | 2018–2019 (n = 74)* | t-test p-value | F-test p-value |
|---|---|---|---|---|
| Night Rotations | 3.7 (0.53) | 3.6 (0.53) | 0.20 | 0.95 |
| ICU Rotations | 3.8 (0.64) | 3.8 (0.74) | 0.87 | 0.20 |
| Jeopardy Rotations | 0.6 (0.57) | 0.5 (0.50) | 0.27 | 0.29 |
| Shift Conflicts | 0.7 (0.90) | 0.3 (0.47) | **< 0.001** | |
| Average Ranking for Assigned Vacations (#, SD) | 1.8 (1.7) | 2.0 (1.4) | 0.52 | |
| Assigned First Choice Rotation (%, SD) | 69.4 (46) | 96.0 (20) | **<0.001** | |

*Values listed as number per resident per year, SD unless otherwise specified.

Comparisons that remain significant after adjustment for multiple comparisons designated with bold text.

**Table 2. Resident schedule quality metrics.**

| Mean (SD) | 2017–2018 (n = 82)* | 2018–2019 (n = 82)* | t-test p-value | F-test p-value |
|---|---|---|---|---|
| Night Rotations | 1.9 (0.84) | 1.9 (0.74) | 0.69 | 0.25 |
| ICU Rotations | 3.2 (0.53) | 3.1 (0.66) | 0.36 | 0.05 |
| Jeopardy Rotations | 0.7 (0.65) | 0.7 (0.50) | 1.00 | 0.02 |
| Shift Conflicts | 0.1 (0.35) | 0.1 (0.23) | 0.13 | |
| Average Ranking for Assigned Vacations (#, SD) | 1.3 (1.2) | 1.6 (1.0) | 0.14 | |
| Assigned First Choice Rotation (%, SD) | 30.5 (47) | 80.5 (41) | <**0.001** | |
| Assigned Second Choice Rotation (%, SD) | 46.3 (50) | 74.4 (44) | <**0.001** | |

*Values listed as number per resident per year, SD unless otherwise specified.

Comparisons that remain significant after adjustment for multiple comparisons designated with bold text.

vacation choices, and assignment of preferred rotations compared to the manual schedule. However, as actual implementation sometimes requires last minute adjustments that may decrease schedule quality, the 2018–2019 schedule may better describe the true net gains of automated schedule generation.

Of 82 residents surveyed, 22 (27%) completed the survey to assess the subjective quality of schedules generated with AIMS. There was a significant increase in both perceived satisfaction and fairness of the schedule with implementation of AIMS (Table 3).

## Discussion

The introduction of AIMS significantly improved both objective and subjective measures of schedule quality by increasing adherence to stated trainee preferences, decreasing transition conflicts, and improving trainee perception of scheduling fairness and quality. This tool significantly increased our ability to schedule trainees for their desired rotations and reduced the number of schedule conflicts generated by overlapping clinical responsibilities, which is anticipated to reduce the strain on the jeopardy pool. Distribution of night and ICU rotations was highly optimized in previous years, as inequity in these rotations was known to cause dissatisfaction. However, AIMS did reduce the variance in jeopardy rotation assignment. No single metric can adequately convey the quality of a schedule, but it is encouraging to note that housestaff perceived the tool to generate a fair schedule and were satisfied with its performance. Perception of fairness and fulfillment of scheduling requests are important aspects of trainee wellness; improved schedules as well as scheduling transparency may potentially lead to decreases in resident burnout. Lack of control over job schedules precipitates work-family conflict and may drive dissatisfaction [22], and increased control over scheduling has been cited in focus groups as a method to reduce burnout [23]. The use of technology to optimize scheduling provides an attractive target to improve well-being without altering duty hours, given the ongoing debate about ideal resident shift length [24].

AIMS features several attributes that make it a valuable tool for chief residents, who are required to create complex schedules, often with little or no prior experience. Once rotation structures and scheduling rules are programmed, they can be reused within an institution

**Table 3. Satisfaction and fairness of the schedule before and after implementation of AIMS.**

| Scale of 1–5; mean (SD) | 2017–2018 | 2018–2019 | t-test p-value |
|---|---|---|---|
| Satisfaction | 3.3 (1.2) | 4.0 (1.1) | **0.048** |
| Fairness | 3.3 (1.4) | 4.2 (1.0) | **0.020** |

until structural changes are made, ensuring consistency over successive academic years. Instead of focusing on minute details of individual schedules, chief residents can evaluate and adjust overarching structural scheduling rules to best suit their trainees. The use of a heuristic algorithm allows schedule generation for even the largest residency programs. The flexibility to hard-code certain elements of the schedule allows more coordination with trainees from other departments which do not use AIMS. The spreadsheet format is easy to manipulate even for those without programming experience. Given the open source nature of AIMS, the structure can be further modified to suit the specific needs of any residency program. Although we did not formally measure chief resident opinion of AIMS, the tool was greatly preferred over manual scheduling, reducing a task that historically took weeks to a matter of days, in keeping with other reports of automation of this onerous task [10, 19].

Our study was limited by the implementation of our tool at a single center, with only two years of scheduling data available for analysis. The heuristic nature of the AIMS algorithm does not ensure an optimal schedule generation. The metrics analyzed were based on prior reports of scheduling quality and chief resident experience and may not reflect all aspects of schedule quality. The survey administered to trainees was not previously validated, and must be interpreted with caution. Trainee opinion about schedule quality for both years was assessed during the 2018–2019 academic year and recall bias may have impacted perception of the previous year's schedule.

## Conclusion

Creating a schedule for internal medicine trainees is challenging, especially in large programs, due to the competing interests of residents, the ACGME, and institutional staffing requirements. Automation has the potential to eliminate error and facilitate the consistency of scheduling between successive years of chief residents. The use of automated scheduling tools such as AIMS can improve metrics of schedule quality, such as avoiding shift conflicts and satisfying more resident preferences. This is reinforced by our survey of trainees, which suggests a subjective improvement in satisfaction and perception of fairness. We have continued to utilize AIMS in our program, a modified version of the algorithm described here was used for the 2019–2020 academic year, and we hope to gain further longitudinal experience with schedule automation. Expansion to other residency programs in our institution may enrich our experience with this tool and confirm a wider applicability. Technologic solutions informed by operations research have the potential to improve the residency experience by granting trainees more control over their time as they learn the practice of medicine.

## Supporting information

**S1 Application. A copy of the scheduling tool used to generate schedules, with step-by-step instructions on setting up the tool for use within other residency programs.** Example information is pre-populated using data from the Yale Internal Medicine Residency Program, with full names of residents censored.
(XLSM)

**S1 Fig. Flowchart illustrating the AIMS algorithm.**
(PPTX)

**S1 Text. Scheduling satisfaction survey.**
(PDF)

**S2 Text. Scheduling satisfaction survey–the list of questions distributed to residents to assess satisfaction and fairness of scheduling with the AIMS tool compared with the**

**manual scheduling process used in years past.**
(DOCX)

## Author Contributions

**Conceptualization:** Frederick M. Howard, Catherine A. Gao.

**Data curation:** Frederick M. Howard.

**Formal analysis:** Frederick M. Howard.

**Investigation:** Frederick M. Howard, Catherine A. Gao.

**Methodology:** Frederick M. Howard, Catherine A. Gao.

**Software:** Frederick M. Howard.

**Supervision:** Christopher Sankey.

**Validation:** Frederick M. Howard.

**Writing – original draft:** Frederick M. Howard, Catherine A. Gao.

**Writing – review & editing:** Catherine A. Gao, Christopher Sankey.

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
