## [Decision Letter · Decision Letter 0]

20 Apr 2020

PONE-D-20-05580

Implementation of an Automated Scheduling Tool Improves Schedule Quality and Resident Satisfaction

PLOS ONE

Dear Dr Gao,

Thank you for submitting your manuscript to PLOS ONE. After careful consideration, we feel that it has merit but does not fully meet PLOS ONE’s publication criteria as it currently stands. Therefore, we invite you to submit a revised version of the manuscript that addresses the points raised during the review process.

We would appreciate receiving your revised manuscript by May 31 2020 11:59PM. To enhance the reproducibility of your results, we recommend that if applicable you deposit your laboratory protocols in protocols.io, where a protocol can be assigned its own identifier (DOI) such that it can be cited independently in the future. For instructions see: http://journals.plos.org/plosone/s/submission-guidelines#loc-laboratory-protocols

We look forward to receiving your revised manuscript.

Kind regards,

Yong-Hong Kuo

Academic Editor

PLOS ONE

Journal Requirements:

2. Your ethics statement must appear in the Methods section of your manuscript. If your ethics statement is written in any section besides the Methods, please move it to the Methods section and delete it from any other section. Please also ensure that your ethics statement is included in your manuscript, as the ethics section of your online submission will not be published alongside your manuscript.

Additional Editor Comments (if provided):

This manuscript has been reviewed by three experts in the area of staff scheduling. Their recommendations and comments tend to be positive. While physician scheduling tool has been studied extensive for many years, they appreciate that the implementation and practical issues discussed in this paper are quite interesting and will be of interested to the reader. They have provided very constructive comments and suggestions for the authors to revise the manuscript. A common comment is that the review of literature and existing work is rather short. Since there have been many studies on this topic, the authors may wish to discuss in more detail the related work. There are a few studies suggested by the reviewers to discuss. The following papers may also be useful for the authors:

Damcı-Kurt, P., Zhang, M., Marentay, B., & Govind, N. (2019). Improving physician schedules by leveraging equalization: Cases from hospitals in US. Omega, 85, 182-193.

Gross, C. N., Brunner, J. O., & Blobner, M. (2019). Hospital physicians can’t get no long-term satisfaction–an indicator for fairness in preference fulfillment on duty schedules. Health care management science, 22(4), 691-708.

Gross, C. N., Fügener, A., & Brunner, J. O. (2018). Online rescheduling of physicians in hospitals. Flexible Services and Manufacturing Journal, 30(1-2), 296-328.

Hong, Y. C., Cohn, A., Epelman, M. A., & Alpert, A. (2019). Creating resident shift schedules under multiple objectives by generating and evaluating the Pareto frontier. Operations Research for Health Care, 23, 100170.

Kuo, Y. H. (2014). Integrating simulation with simulated annealing for scheduling physicians in an understaffed emergency department. HKIE Transactions, 21(4), 253-261.

Schoenfelder, J., & Pfefferlen, C. (2018). Decision support for the physician scheduling process at a German hospital. Service Science, 10(3), 215-229.

Tohidi, M., Kazemi Zanjani, M., & Contreras, I. (2019). Integrated physician and clinic scheduling in ambulatory polyclinics. Journal of the Operational Research Society, 70(2), 177-191.

Vermuyten, H., Rosa, J. N., Marques, I., Belien, J., & Barbosa-Póvoa, A. (2018). Integrated staff scheduling at a medical emergency service: An optimisation approach. Expert Systems with Applications, 112, 62-76.

Please seriously address the reviewers' concerns. Unsuccessful revision can lead to rejection of the paper.

Reviewers' comments:

Reviewer's Responses to Questions

**Comments to the Author**

1. Is the manuscript technically sound, and do the data support the conclusions?

Reviewer #1: Yes

Reviewer #2: Yes

Reviewer #3: Partly

2. Has the statistical analysis been performed appropriately and rigorously? 

Reviewer #1: Yes

Reviewer #2: Yes

Reviewer #3: Yes

3. Have the authors made all data underlying the findings in their manuscript fully available?

Reviewer #1: Yes

Reviewer #2: Yes

Reviewer #3: No

4. Is the manuscript presented in an intelligible fashion and written in standard English?

Reviewer #1: Yes

Reviewer #2: Yes

Reviewer #3: Yes

5. Review Comments to the Author

Reviewer #1: This work describes the development and implementation of a heuristic-based scheduling tool applied to a large internal medicine residency program including before and after metrics. This is a pragmatic description of implementation and analysis, including the real-world accommodations made for practicality (e.g. essential elements manually entered in to AIMS tool prior to automated scheduling). The core functionality is automated sequential scheduling lottery. Qualitative and quantitative outcomes were assessed and appropriate statistical analysis was applied.

In particular, the authors provide the appropriate level of detail to both understand their methodology but maintain a level of accessibility for the target audience of whom many will not be familiar with automated scheduling heuristics. In addition, inclusion of both qualitative and quantitative measures of success is of particular importance for scheduling outcomes. The supplemental materials will be a valuable resource for other programs to implement their strategy.

Specifically:

1. Results: With the Jeopardy and night rotations, is there any variability in day night transitions or will a set number of night float shifts always occur for a block without potential for night coverage elsewhere? If there is any heterogeneity in this, consider including a metric of day to night transitions to compare automated to manual scheduling.

2. Lines 171-173: Discussion of wellness as it related to scheduling satisfaction is appropriate. Consider expanding this area of the discussion more as this strongly underscores the role technologic solutions may play in decreasing provider burnout and improving workplace wellness.

3. Lines 174-179: These are all valuable attributes as described. In addition, given that the individual preparing the schedule is likely to be different in consecutive years, consider including reflection on the role of technological solutions to decrease the “learning curve” for schedule development.

4. Lines 180-186: Consider including reflection on how the survey instrument itself—while appropriate—was not previously validated.

The authors should be commended for their enthusiasm for this topic and their desire to apply technological solutions to common training problems with scheduling.

Reviewer #2: Manuscript Number: PONE-D-20-05580

Referee Report on the paper “Implementation of an Automated Scheduling Tool Improves Schedule Quality and Resident Satisfaction” submitted to PLOS One

Summary and recommendation

This referee report discusses an article about the quality of automating rotation schedules for residents. The authors developed a rotation scheduling tool that was implemented for the Yale New Haven Hospital Internal Medicine Residency Program. In their study, they compare the quality of schedules before and after implementation of the new tool based on objective as well as subjective perception.

Even though the methodical contribution is minor (and not necessary for the journal submitted), the implementation and analysis in a working environment are quite interesting. In general, the paper is very well structured and well written. However, the paper misses a careful literature review. Additionally, the description of the tool/algorithm is only superficial. Based on the following comments, we recommend a major revision.

Major comments

Introduction: The authors jump directly into the topic. Even if you are somewhat familiar with the subject, essential terms should be introduced for the reader (e.g. rotation, jeopardy rotation etc.). A clear problem description might help as well. Please note that residency programs are quite different in various countries. The literature review from line 57-69 is rather short and misses some essential papers in resident scheduling literature (at least from a methodological point of view). We would recommend the literature review “State of the art in physician scheduling” by Erhard et al. (2018) as a starting point for the review. Forward and backward search might be very helpful finding relevant papers.

Method: The description of the scheduling tool is lagging. The process is not entirely transparent at first glance. For instance, you could add a pseudocode to the appendix or use a flow chart for the process. We have a couple of open questions that show the ambiguity or incompleteness of the text.

- What is the time horizon of your schedule?

- Are the problems for residents and interns independent, i.e. you solve both independently? If not, then some explanation in the text is needed.

- What do you mean by scheduling lottery? How often is this lottery performed within the time horizon?

- What kind of preferences do you look at, i.e., only rotation preferences or overnight duties?

- Can this lottery be manipulated, i.e., when I know that I got my first preference in the last lottery, can I change my preferences so that a lower priority might be my true first one? Or are you running the lottery for each 2-week horizon using the data collected for the whole year? It is somewhat unclear in the text.

- What is the “jeopardy” pool? Please explain and add a description (see comment our comments to the introduction).

- A visualization of the algorithm might be helpful.

- What is the objective function of your algorithm, or do you just construct a feasible solution? We assume the latter but it is not clear in the text.

- What are your objective metrics based on, i.e., expert knowledge or literature? We assume the former but again it is not clear in the text. A motivation based on literature might be valuable as well.

- On page 4 you are fixing some input values. What is the effect of relaxing this assumption? We guess you achieve better schedules? But understand that e.g. external rotations are fixed. However, are you planning rotations that are external rotations for other units? Then some coordination between units is necessary. You should motivate the assumption.

- On your tally, what happens if you do not grant any request? We mean: If nothing is fulfilled, then a resident/intern has 0 and is never chosen? How do you initialize the heuristic?

- What happens in case you have less residents/interns available than needed?

- Are you limiting the number of requests of any kind (e.g. preferences, vacations) for each resident/intern?

Results:

We like the comparison between the two years which can be seen as a contribution for itself. However, it might be biased. Just as an idea, you might regenerate schedules for 2017-2018 retrospective and compare the results with the realized ones. Also, it was unclear whether you base your evaluations on the planned or realized (with re-planning) schedules. We think and hope you use planned schedules. If so, the subjective assessment might be biased by re-planning as well. Please make it clear and comment on it in the text. In future, we recommend detailing the subjective questionnaire in alignment with the quality metrics. Can you say something about the savings for the chief resident? E.g. is (s)he faster on top of the quality gains? Also, will the procedure be used next year as well. If so, why? If not so, why? A short outlook might be useful for other practitioners. Just a note from methodological point of view, are you correcting multiple hypotheses testing, or are you just performing single tests in your analysis? For the latter, some of your results might not be significant anymore. You might use a step-wise procedure and most of your findings should be the same.

Minor comments

- The authors introduce some abbreviations twice, i.e., AIMS and ACGME. You might don’t want to use abbreviations in the abstract.

- The conclusion is very short. You might want to address aspects you have not considered but are from interest.

We hope the authors regard our feedback as helpful for improving the paper. Furthermore, we suggest extending your team by a researcher more familiar with operations research to improve your scheduling part. We are pretty sure that some qualified persons exist at your university (maybe in industrial engineering or business). The report can be found in the attachment as well.

Reviewer #3: The paper describes the implementation of a resident scheduling algorithm at a large hospital. While this is not really new, I really liked the link to a survey that demonstrates an increase in satisfaction after the new software. I think the paper should be published, but ask for two things:

1) Literature: I think you should discuss a little what has been done. There is a relatively recent literature review on physician scheduling (Erhard et al. 2018) that you should have a look at, and probably refer your readers to. I am personally aware of some related papers discussing new approaches in physician scheduling along a real-life implementation and comparison of results, such as Bowers et al. 2016 comparing an equital and a preferences-orientated scheduling, or Fügener et al. 2015 who include both fairness concerns and individual preferences in physician scheduling. Erhard et al. 2018 provide a list of papers with respect to, e.g., residents, fairness, preferences. Again, there is no need to write an extensive literature review, but at least show what comparable studies exist, and maybe refer to the review paper.

2) Algorithm: You should discuss more clearly how your approach works - it could be an appendix if it takes too much space.

I am happy to review a revised version.

References:

Bowers, M. R., Noon, C. E., Wu, W., & Bass, J. K. (2016). Neonatal physician scheduling at the University of Tennessee Medical Center. Interfaces, 46(2), 168-182.

Erhard, M., Schoenfelder, J., Fügener, A., & Brunner, J. O. (2018). State of the art in physician scheduling. European Journal of Operational Research, 265(1), 1-18.

6. PLOS authors have the option to publish the peer review history of their article (what does this mean?). If published, this will include your full peer review and any attached files.

Reviewer #1: Yes: Stefanie J. Hollenbach, M.D., M.S.

Reviewer #2: Yes: Jens O. Brunner

Reviewer #3: No

---

## [Author Response · Author response to Decision Letter 0]

22 May 2020

To begin, we would like to comment on some updates to our schedule analysis. In response to reviewer 2, we regenerated the 2017-2018 schedule to ensure year to year variability did not lead to the superior schedule generation with our tool. When regenerating this schedule, we carefully ensured that the same number of rotations / rotation types were covered. When doing this, we discovered an inaccuracy with how the number of nights and ICU rotations were calculated per resident; specifically in the intern year. This resulted in one intern with an additional night rotation, and four interns with an additional ICU rotation. We have since updated the schedule. We were previously using the number of night / ICU rotations that were reported by the authors who manually generated the prior schedule – highlighting how miscalculations can occur with manual schedule work.

Second – we have updated our calculations for shift conflicts – due to overcounting the number of conflicts that occurred. This has increased the significance of one of our comparisons, and made one of the comparisons no longer significant – we feel the provided number of conflicts is a more accurate measure of the number of times a resident would need to be pulled to provide coverage.

Finally, we have corrected the standard deviation for the average ranking of the assigned vacation choices. Each resident gets two vacation choices, and the average rank they had assigned to each vacation choice is reported. The standard deviation was previously calculated from the summed rank of vacation choices, not the average – the standard deviation of the average is effectively half.

Comments from Reviewer 1:

 1. Results: With the Jeopardy and night rotations, is there any variability in day night transitions or will a set number of night float shifts always occur for a block without potential for night coverage elsewhere? If there is any heterogeneity in this, consider including a metric of day to night transitions to compare automated to manual scheduling. 

There is indeed a slight variability in number of night-to-day transitions per block; our program mandates that residents spend no more than 2 weeks on a night rotation, so one easily-obtainable metric of night to day transitions is the number of night rotations per block. We have provided this metric for each year analyzed as a supplement.

2. Lines 171-173: Discussion of wellness as it related to scheduling satisfaction is appropriate. Consider expanding this area of the discussion more as this strongly underscores the role technologic solutions may play in decreasing provider burnout and improving workplace wellness.

We agree that technologic advances – in scheduling and in other avenues – have the potential to improve resident wellness, and have further discussed the impact of our work in this regard.

 3. Lines 174-179: These are all valuable attributes as described. In addition, given that the individual preparing the schedule is likely to be different in consecutive years, consider including reflection on the role of technological solutions to decrease the “learning curve” for schedule development.

We appreciate the invitation to provide our reflection on the role of automated scheduling solutions on the learning curve for chief residents, and have detailed our opinions as such.

 4. Lines 180-186: Consider including reflection on how the survey instrument itself—while appropriate—was not previously validated.

We have called attention to the lack of validation of the survey in the discussion.

Comments from Reviewer 2:

Major comments Introduction: The authors jump directly into the topic. Even if you are somewhat familiar with the subject, essential terms should be introduced for the reader (e.g. rotation, jeopardy rotation etc.). A clear problem description might help as well. Please note that residency programs are quite different in various countries. The literature review from line 57-69 is rather short and misses some essential papers in resident scheduling literature (at least from a methodological point of view). We would recommend the literature review “State of the art in physician scheduling” by Erhard et al. (2018) as a starting point for the review. Forward and backward search might be very helpful finding relevant papers.

Thank you for calling attention to the ambiguous terminology; we have provided some further clarification of these terms and further introduction to the topic as a whole. 

We have also utilized the review recommended as a starting point to expand our literature review.   Method: The description of the scheduling tool is lagging. The process is not entirely transparent at first glance. For instance, you could add a pseudocode to the appendix or use a flow chart for the process. We have a couple of open questions that show the ambiguity or incompleteness of the text.  - What is the time horizon of your schedule?

We have clarified the time horizon (1 year at a time) in the first paragraph of our methods.

 - Are the problems for residents and interns independent, i.e. you solve both independently? If not, then some explanation in the text is needed.

Both problems are solved concurrently. We have detailed the methodology further with a flowchart that may better explain this. Essentially each resident is assigned a label – such as ‘intern’, ‘psychiatry’, ‘senior resident’, etc, that designates which rotations can be assigned to which residents.

 - What do you mean by scheduling lottery? How often is this lottery performed within the time horizon?

The initial assignments of vacations are done by lottery – i.e. a resident is selected and then given their first choice of vacation if it is available. The order of selection for subsequent lotteries is influenced by the tally of previously satisfied preferences.

 - What kind of preferences do you look at, i.e., only rotation preferences or overnight duties? The only resident input provided is vacation and rotation preferences.

- Can this lottery be manipulated, i.e., when I know that I got my first preference in the last lottery, can I change my preferences so that a lower priority might be my true first one? Or are you running the lottery for each 2-week horizon using the data collected for the whole year? It is somewhat unclear in the text.

The lottery cannot be manipulated, as all preferences are submitted upfront and then the lotteries are run sequentially. 

 - What is the “jeopardy” pool? Please explain and add a description (see comment our comments to the introduction).

We have concretely defined this term in the introduction to hopefully remove all ambiguity.

 - A visualization of the algorithm might be helpful.

We have created a flow chart to illustrate the algorithm

 - What is the objective function of your algorithm, or do you just construct a feasible solution? We assume the latter but it is not clear in the text.

Excellent question – an in-process version of our tool (which we used for the 2019-2020 schedule) uses an objective function with iterative schedule generation to further optimize the results, but in this version a single solution is generated at a time. We have clarified in the text.

 - What are your objective metrics based on, i.e., expert knowledge or literature? We assume the former but again it is not clear in the text. A motivation based on literature might be valuable as well.

You are correct that we selected these metrics based on expert knowledge – but we find recurring themes in the literature to support our selection, although no universally standard metric exists given the differing needs of training programs. We have added a discussion on the motivation for these metrics.

 - On page 4 you are fixing some input values. What is the effect of relaxing this assumption? We guess you achieve better schedules? But understand that e.g. external rotations are fixed. However, are you planning rotations that are external rotations for other units? Then some coordination between units is necessary. You should motivate the assumption.

Relaxing the fixed input would likely provide better schedules, but would require participation from the program director coordinating rotating residents from other services (i.e. emergency medicine, primary care, etc). Currently, we are provided with a list of rotators from these other services, and this causes fluctuations in the number of residents required from our program for each block. We have attempted to clarify this in the text.

 - On your tally, what happens if you do not grant any request? We mean: If nothing is fulfilled, then a resident/intern has 0 and is never chosen? How do you initialize the heuristic?

We have clarified this further in the text – the tally is initialized at zero, so every resident has an equal chance of being chosen for their first vacation. Each resident will be mandated to receive two vacations, and have ranked all blocks in order of preference, so if they receive their last choice vacation, the tally could increase by 14.

 - What happens in case you have less residents/interns available than needed?

We are lucky to have enough redundancy to provide coverage to all required services.

- Are you limiting the number of requests of any kind (e.g. preferences, vacations) for each resident/intern?

Each resident/intern ranks every potential vacation choice, so we get a complete ordering of their preferences. They also rank their top three rotation preferences. We have clarified as such in the text. We did allow free text specification of any other considerations, such as weddings or religious holidays.

  Results: We like the comparison between the two years which can be seen as a contribution for itself. However, it might be biased. Just as an idea, you might regenerate schedules for 2017-2018 retrospective and compare the results with the realized ones. 

We have provided this comparison as a supplement as an exploratory analysis, although there are factors aside from resident preferences (such as specific requests for weddings or other important events) that are unavailable for the 2017-2018 schedule that may provide our regenerated schedule with more flexibility and thus introduce more bias. Additionally, replanning to accommodate last minute changes can potentially reduce other metrics of schedule quality when looking at the actually implemented schedule. Thus, this regeneration may provide an overoptimistic view of actual schedule quality obtained through automation.

Also, it was unclear whether you base your evaluations on the planned or realized (with re-planning) schedules. We think and hope you use planned schedules. If so, the subjective assessment might be biased by re-planning as well. Please make it clear and comment on it in the text. 

The subjective assessment of schedule fairness and satisfaction were based on realized schedules, which indeed may have introduced bias. However, the perception of fairness and satisfaction with schedule quality were ‘lived’ experiences, and we did not feel it would be accurate to have residents compare an abstract schedule generated by AIMS to their lived experiences with the manual 2018-2019 

In future, we recommend detailing the subjective questionnaire in alignment with the quality metrics. Can you say something about the savings for the chief resident? E.g. is (s)he faster on top of the quality gains? 

We have added in the discussion a subjective assessment of time saved, as a formal assessment of the time difference was not performed. When taking on this task from the previous years’ chiefs, they unanimous rated scheduling as the most challenging aspect of the job, and we can state with confidence the task was much more enjoyable this year.

Also, will the procedure be used next year as well. If so, why? If not so, why? A short outlook might be useful for other practitioners. 

We did use this procedure for the 2019 – 2020 academic year, with a rewritten tool and algorithmic modifications.

Just a note from methodological point of view, are you correcting multiple hypotheses testing, or are you just performing single tests in your analysis? For the latter, some of your results might not be significant anymore. You might use a step-wise procedure and most of your findings should be the same.

Thank you for raising this point, we have provided a correction for multiple hypothesis testing and specified this in our methodology.   Minor comments - The authors introduce some abbreviations twice, i.e., AIMS and ACGME. You might don’t want to use abbreviations in the abstract.

Thank you for noticing these duplicates, we have removed them.

 - The conclusion is very short. You might want to address aspects you have not considered but are from interest.

Thank you for the comment – we have added to our conclusion, and moved some items from discussion better suited for conclusion to that section.

Reviewer #3: The paper describes the implementation of a resident scheduling algorithm at a large hospital. While this is not really new, I really liked the link to a survey that demonstrates an increase in satisfaction after the new software. I think the paper should be published, but ask for two things: 1) Literature: I think you should discuss a little what has been done. There is a relatively recent literature review on physician scheduling (Erhard et al. 2018) that you should have a look at, and probably refer your readers to. I am personally aware of some related papers discussing new approaches in physician scheduling along a real-life implementation and comparison of results, such as Bowers et al. 2016 comparing an equital and a preferences-orientated scheduling, or Fügener et al. 2015 who include both fairness concerns and individual preferences in physician scheduling. Erhard et al. 2018 provide a list of papers with respect to, e.g., residents, fairness, preferences. Again, there is no need to write an extensive literature review, but at least show what comparable studies exist, and maybe refer to the review paper. 

Thank you for this comment. We have expanded on our literature review, referred our readers to Erhard et al, and included a selection of references in our background discussion, in particular studies addressing residency scheduling. 

 2) Algorithm: You should discuss more clearly how your approach works - it could be an appendix if it takes too much space.

Thank you, we have included a more thorough description of the algorithm as a flow chart to illustrate the steps taken.

---

## [Decision Letter · Decision Letter 1]

13 Jul 2020

PONE-D-20-05580R1

Implementation of an Automated Scheduling Tool Improves Schedule Quality and Resident Satisfaction

PLOS ONE

Dear Dr. Gao,

Thank you for submitting your manuscript to PLOS ONE. The revision has been reviewed by two reviewers from the last round. They are satisfied with the revision and have favourable recommendations. There are some minor suggestions for the authors to consider for the final manuscript.

We look forward to receiving your revised manuscript.

Kind regards,

Yong-Hong Kuo

Academic Editor

PLOS ONE

Reviewers' comments:

Reviewer's Responses to Questions

**Comments to the Author**

1. If the authors have adequately addressed your comments raised in a previous round of review and you feel that this manuscript is now acceptable for publication, you may indicate that here to bypass the “Comments to the Author” section, enter your conflict of interest statement in the “Confidential to Editor” section, and submit your "Accept" recommendation.

Reviewer #2: All comments have been addressed

Reviewer #3: All comments have been addressed

2. Is the manuscript technically sound, and do the data support the conclusions?

Reviewer #2: Yes

Reviewer #3: Yes

3. Has the statistical analysis been performed appropriately and rigorously? 

Reviewer #2: Yes

Reviewer #3: Yes

4. Have the authors made all data underlying the findings in their manuscript fully available?

Reviewer #2: Yes

Reviewer #3: Yes

5. Is the manuscript presented in an intelligible fashion and written in standard English?

Reviewer #2: Yes

Reviewer #3: Yes

6. Review Comments to the Author

Reviewer #2: We think the authors revised their paper very well. Our comments have all been taken into account. The revision of the literature has been very successful, and the paper can now generally be read more smoothly. So in our view, there is nothing to be said against accepting the paper for publication.

Minor comments

- p. 5, line 89: You should use “constraints” rather than “restraints”.

- p. 6, line 103: You might want to use “trainee” rather than “learner”.

Reviewer #3: The autors improved the manuscript. I like the revision, and have only a minor comment.

I would rather have the reference within the sentence it belongs to, e.g., "...pathway to independent practive [1]." instead of "...independent practive. [1]". However, please leave a space before the reference (unlike e.g., on page 9, references [20] and [9].

Thanks for the opportunity to review your paper!

7. PLOS authors have the option to publish the peer review history of their article (what does this mean?). If published, this will include your full peer review and any attached files.

Reviewer #2: **Yes: **Jens O. Brunner

Reviewer #3: No

---

## [Author Response · Author response to Decision Letter 1]

14 Jul 2020

Response to reviewers:

Reviewer #2: We think the authors revised their paper very well. Our comments have all been taken into account. The revision of the literature has been very successful, and the paper can now generally be read more smoothly. So in our view, there is nothing to be said against accepting the paper for publication.  Minor comments - p. 5, line 89: You should use “constraints” rather than “restraints”. - p. 6, line 103: You might want to use “trainee” rather than “learner”.

Response: Thank you for taking the time to review our paper and for your comments; we have made the recommended changes. 

Reviewer #3: The autors improved the manuscript. I like the revision, and have only a minor comment.  I would rather have the reference within the sentence it belongs to, e.g., "...pathway to independent practive [1]." instead of "...independent practive. [1]". However, please leave a space before the reference (unlike e.g., on page 9, references [20] and [9].  Thanks for the opportunity to review your paper!

Response: Thank you for taking the time to review our paper and for your comments; we have made the recommended changes in citation formatting.

---

## [Editor Report · Decision Letter 2]

17 Jul 2020

Implementation of an Automated Scheduling Tool Improves Schedule Quality and Resident Satisfaction

PONE-D-20-05580R2

Dear Dr. Gao,

We’re pleased to inform you that your manuscript has been judged scientifically suitable for publication and will be formally accepted for publication once it meets all outstanding technical requirements.

Kind regards,

Yong-Hong Kuo

Academic Editor

PLOS ONE

Additional Editor Comments (optional):

The authors have successfully addressed the reviewers' concerns. Thus, I recommend acceptance of the work.
---

## [Editor Report · Acceptance letter]

29 Jul 2020

PONE-D-20-05580R2 

Implementation of an Automated Scheduling Tool Improves Schedule Quality and Resident Satisfaction 

Dear Dr. Gao:

I'm pleased to inform you that your manuscript has been deemed suitable for publication in PLOS ONE. Congratulations! Your manuscript is now with our production department. 

Kind regards, 

on behalf of

Dr. Yong-Hong Kuo 

Academic Editor

PLOS ONE